# BRAIN: BEHAVIORAL RESPONSES AND ARTIFICIAL INTELLIGENCE NEURAL-MODELING FOR CONSUMER DECISION-MAKING

## ABSTRACT

This research investigates consumer neuroscience and neuromarketing through a multivariate methodology, employing Principal Component Analysis (PCA) and deep learning neural networks to interpret consumer responses to functional products. EEG signals were collected, recorded, and analyzed from 16 individuals aged 20 to 29 to identify significant neuronal markers related to consumer choices. The pivotal factors influencing decision-making were identified as the low beta and low gamma frequency bands, as well as participants' attention and meditation levels. The findings validate the effectiveness of our approach, demonstrating its applicability across various fields requiring accurate and reliable classification. Additionally, it is recommended to explore the potential applications of this study in the food industry by creating personalized nutrition strategies based on individuals' brain activity patterns.

**Keywords:** Decision-Making, PCA, DCNN and Neuromarketing.

## 1 INTRODUCTION

Understanding factors influencing consumer purchasing decisions interests both scholars and industry. Marketing departments strive to grasp consumer preferences to improve advertising efficiency and reduce costs. The Mexican food sector, contributing 3.9% to GDP in 2020 [5], is socially, culturally, and economically significant. Latin America's obesity and overweight rates rose by 58%, with Mexico showing 76.4% in adults and 35.6% in children aged 5-11 years [19]. Unhealthy diets in Mexico contribute to these health issues [6], linking to chronic diseases like hypertension and type 2 diabetes. This situation urges global consumers to adopt healthier diets. Thus, the food industry must consider consumer demands to develop strategies that promote both commercial success and consumer well-being. In 2023, Mexico dropped from second to fifth in the global obesity ranking, per ENSANUT 2020, due to joint public health efforts. Global anti-obesity measures include calorie and fat content labeling [17], enabling consumers to evaluate nutritional value [7], and a tax on sugary drinks.

However, more research is needed to fully understand policy impacts on consumer behavior due to limited data. The shift in consumer attitudes has opened a market for beneficial nutrition products like functional foods. Although functional foods are related to high health benefits, consumers do not tend to accept them. So, you must recognize the emotional connection to figure out their actual tastes and see if they actually like a product. This would, thus, allow companies that manufacture these foods to create long-term and effective initiatives based on concrete data which will allow for the successful adoption of alternative proteins in the marketplace. Wherein the taste is more the sense among his emotions when it comes to preference or aversion. Taste buds give us the ability to tell what is sweet, salty, sour, and bitter; that is why I talk about taste as a whole here: because while tasting can tell you if you generally like something or not it cannot help when we are differentiating between complex flavors [22]. While other senses like smell act on the perception of flavor as well, taste is the most important since although our sense of smell can bias us into thinking we are tasting something else by mixing stimuli. The taste bud receptors send signals to the brain when foods or liquids make contact with them, and that is what makes a flavor sensation. This can be both individual and dependent upon exposure and experience, which can explain how our taste for certain foods identified as unpalatable can often be reappraised over time. This is never so relevant as for functional

foods, which sens experience high levels of neophobia (unknown flavours are avoided in a manner similar to the non-registration of colours by humans or identification networks) [3. Companies have demonstrated how the use of Neuromarketing techniques has helped to better understand consumer reactions to their products.

More specifically for companies that generate beneficial health foods, they have helped to have greater acceptance and positive perception of the brand, increasing acceptance of a healthy product and changing consumers' perception of a functional food. This is achieved through tools that help Neuromarketing to identify which product characteristics can positively influence the choice of healthy options. In order to have greater insight into the knowledge of decision making, there are studies that analyze the prefrontal cortex studies. Neuromarketing tools such as EEG and functional magnetic resonance imaging have been useful to understand purchase decisions. Studies show that PCA is the preferred method for analyzing organoleptic properties. [1; 16; 14; 9] Consumers are influenced by various stimuli that evoke different motivations, emotions, and responses. Companies must delve deeper into consumer behavior for effective, sustainable marketing tactics. The document is organized as follows: Section 2 outlines our EEG signal collection methodology and theoretical background, including PCA and its application in the analysis of brain wave patterns. Section 3 explores the main findings of brain activity in product evaluations that improve deep convolutional neural network training and compares with other models. Section 4 summarizes key insights, future research avenues, and study conclusions.

## 2 METHOD

### 2.1 GENERAL METHODOLOGY

Consumer behavior considers the psychological and physical aspects of factors like hunger and health, motivation, and emotions. This information can assist companies in creating products and marketing strategies that work best. The choice of the sample took into account obesity, eating habits and media exposure. In the next phase of research, participants' biological responses to physiological activities including eating, sleep and stress will be tracked. **BRAIN** stands for Behavioral Responses and Artificial Intelligence Neural-modeling and it is an acronym that directly highlights the objective of the study, which is to model behavioral responses through neural networks based on artificial intelligence. In addition, the use of this acronym reflects the focus on brain activity (EEG) and its application in consumer behavior analysis. Thus, **BRAIN** is a neural network trained with images of a corpus, and its output is weighted only by the most relevant EEG signals activated in the consumer's brain. Figure 1(a) depicts the general model of the proposed system, which can be generalized into three parts: i) the inputs of the model, which are images and EEG time series, ii) the outputs, which are the prediction of whether the consumer likes the functional product or not, and iii) the feedback, which will help us measure the system's efficiency through a confusion matrix and an ROC curve. From Figure 1, this study first recorded the brain activity of 16 participants who collected EEG samples. A total of 124 EEG tests were collected, recorded and analyzed to uncover patterns linked to specific stimuli. In addition, the panelists rated eight different functional products, with 1,291 photos of the samples analyzed for preferences. Facial expressions (1,330 in total) were also evaluated to determine reactions to taste samples. The positive findings could indicate the link between brain activity and perceptions of functional products consumed. The database can be downloaded at `https://acortar.link/IYyMyV`.

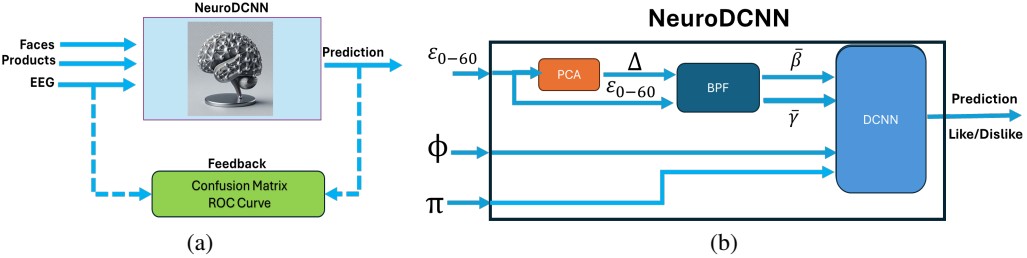

Figure 1: Schemes of the proposed model, (a) General Model and (b) Extended Model.

Figure 1(b) depicts the extended model of the proposal, which is divided into three main parts: i) Principal Component Analysis (PCA) of EEG signals ($\varepsilon_{0-60}$) from 0.5 to 60 Hz, ii) Band-pass Filtering (*BPF*) of the main rhythms present in consumer decision-making ($\Delta$), and iii) Training of a Deep Learning Convolutional Neural Network with the image corpus of faces ($\Phi$) along with products ($\pi$) weighted with low beta ($\overline{\beta}$) and low gamma ($\overline{\gamma}$) signals. Thus, the output of the system is whether a functional product is liked or not, as in the general model. Finally, the efficiency of *BRAIN* is measured using the Confusion Matrix (CM) and the receiver operating characteristic curve (ROC curve).

## 2.2 EXPERIMENTAL METHODOLOGY

In this section, we describe the methodology used to measure the brain response of the panelists while tasting eight samples of functional products containing the four-basic taste sensations: sour, bitter, salty, and sweet. A laptop model Machenike T58 and the COLAB Pro platform code were used to analyze the data. COLAB runs on the Google Compute Engine backend in Python 3, utilizing system RAM 51.0 GB, a T4 GPU with RAM of 15.0 GB, and disk space 201.2 GB. EEG was applied as a neuroscientific technique in Phase 1 of an ongoing study on the use of neuromarketing to understand consumer behavior. This phase started with getting the panelists ready to collect initial data on functional product preferences: Environment, schedule and time intervals - Sensory Evaluation Manual: Sensory evaluation methods [25], as well as how to select a panel, organize a test and what the results are telling you is described in this no-nonsense manual. Sensory evaluation was originally developed during World War II to understand why soldiers rejected food, yet due to the sensitivity of human senses, remains a valuable tool even when instrumental methods exist to determine quality.

This article reports results from Phase 1 and evaluates whether or not to proceed with the research. Phase 1 was divided as follows: i) Elicit preferences beliefs of the panelists, ii) Collect EEG activity in participants brains, iii) Validation information and iv) Discover connected brain waves. With this information, large-scale replicability in the future becomes possible. Phase 2 will explore the combination of basic taste senses, as after Phase 1. Phase 1 will be run under controlled conditions for a minimum of five days per flavour, before combining flavours to eliminate response bias. We recruited sixteen participants and asked them to log their likes and dislikes through an application when tasting functional food for the first time. Regarding the ethics of the tests carried out in the experiment, an exhaustive review of the study process was carried out, which ranges from the beginning of planning the test to be carried out to the explicit explanation of the analysis of the results, ensuring that at no time were puts the safety or well-being of participants at risk. This study was reviewed and approved by an Institutional Review Board (IRB) made up of members of the institution where the test was carried out, from students, teachers and administrators who agreed that the objective of guaranteeing compliance was achieved. of the highest ethical standards. The research subjects had complete freedom to decide whether or not they wanted to participate in the test, and they were informed in a clear and detailed manner about each of the aspects that their participation involved. Participants were fully aware that images of their face, their responses and EEG shots of their brain were and will be used, understanding the purpose and use of this data in the research. Express written consent was obtained from each participant, in a clear and very detailed manner explaining the objectives and fines, where they were informed of the risks and benefits of the experiment and, above all, how the data provided would be used and continued. Through a filling out sheet, the voluntary nature of their participation was highlighted, the authorization of the use of their information, that they were in total agreement to participate in the experiment, where they also indicated that they were fully aware and understood of the process to be carried out. and that they had the right to withdraw at any time without adverse consequences. It is worth mentioning that only one person on the team was in charge of being in contact with this data during the execution of the experiment and that same person was in charge of encrypting the personal data of the participants. This person was previously trained and indicated on each sheet filled out by the participant that they were committed to making correct use of the information they observed. For the future continuity of the research, it has been considered not to obtain name data, but only physiological data such as age and gender. For each panelist, two samples of each basic taste sensation were provided and asked to indicate if they liked or disliked each sample. It should be noted that one of the reasons why the sample was limited to 16 participants derives from the provisions of the Sensory Evaluation Manual [25], where the panel size is considered to be a minimum of 8 people. Results are generally better with a small, well-trained team than with a large, untrained team. Therefore, when considering

two aspects to analyze: i) taste or dislike and ii) flavors, the sample was doubled. The responses of the panelists were compared with the EEG data, leaving a gap between each sample to taste an unsalted cookie and a sip of water to remove any taste detected before by the panelist.

In a study involving 16 panelists aged 20 to 29 years (median = 25), brain activity and facial expressions were recorded using EEG while tasting samples. The study, which ensured prior consent, consisted of 43. 7% women and 56. 3% men. The analysis of these recordings aimed to assess the correlation between panelist responses and brain activity measurements, offering valuable insights for the food industry in product development.

## 2.3 ELECTROENCEPHALOGRAPHIC SIGNALS

### 2.3.1 ACQUISITION

An EEG signal captures electrical currents from activated neurons in the cerebral cortex, detectable by EEG systems. These systems function as brain-computer interfaces (BCI), using surface electrodes placed according to the International 10-20 Positioning System, [2; 8]. The ThinkGear TGAM1 IS a non-invasive BCI that records and analyzes neural signals, achieving a precision level of 98% [13]. This is possible taking into account the data integrity, discarding those samples where the flag or indicator called Poor Quality Value is equal to zero. The attributes of brain wave activity monitored by the sensor are quantified through designated flag values: Signal Flatness (25), Signal Excessiveness (26), Power Ratio (27), and Off-Head Detection (29). Concurrent flag indications are possible; for example, a flag value of 51 for suboptimal signal quality indicates non-compliance with both the flatness and excessiveness criteria. Flag values have been meticulously selected to ensure the uniqueness of each possible combination. In addition, a flag value of 200 is indicative of a state in which the sensor has recorded off-head conditions for a duration of four seconds. The RAW signal of the time series with a sampling frequency of 512 Hz, which allows sampling up to 256 Hz, thus making it possible to separate the signals into fundamental brain wave bands or rhythms, as well as to eliminate the 60 Hz frequency, due to electromagnetic interference from the AC power line through a 60 Hz $\pm 0.1\%$ Notch filter namely a Band Stop Filter. BCIs, as described in the similar studies as [20; 11; 10; 1] can predict user intention through EEG signals using different approaches techniques of BCI behaviour neuro-siphy research has been utilise in neuromarketing since it aims to analyse brain responses during certain marketing stimuli. It had also compared various feature selection methods to achieve accuracy detection of consumer preferences, and reported the performance of classifiers would have better results with the use of feature selection.

### 2.3.2 TIME SERIES CORPUS

The data set $\varepsilon_{0-60}$, with 124 samples that vary in time, each containing 9,000 to 15,000 data points, is analyzed using the TGAM1 EEG sensor at a rate of 512 samples per second for the *RAW* time series and other brain rhythms at one sample per second. This study focuses on raw EEG signals to extract information from various frequency bands and assesses Attention and Meditation levels during PCA tasks. Figure references Figures 2(a) and 2(b) illustrate brain responses to a product both positively and negatively.

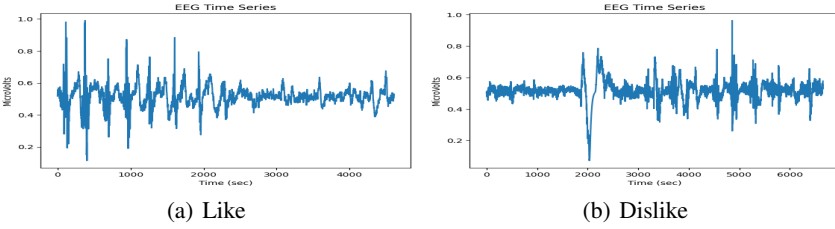

(a) Like            (b) Dislike

Figure 2: EEG samples of Panelists showing their response to something sweet.

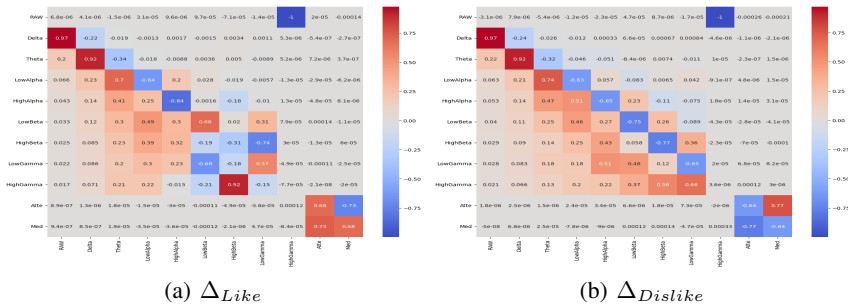

(a) $\Delta_{Like}$        (b) $\Delta_{Dislike}$

Figure 3: Principal Components Analysis ($\Delta$) when the panelists taste a functional product.

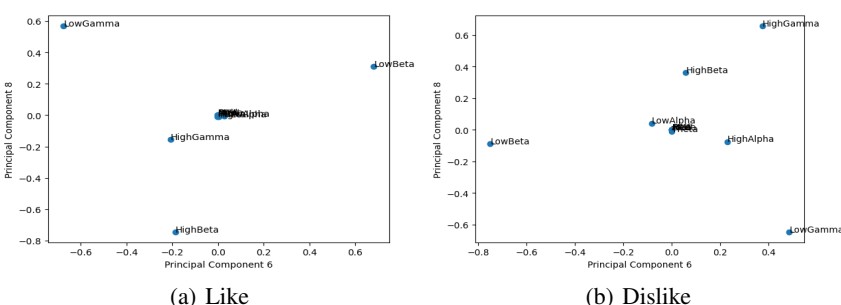

(a) Like        (b) Dislike

Figure 4: Principal Components 6 and 8 ($\Delta$) when the panelists taste a functional product.

### 2.3.3 PRINCIPAL COMPONENT ANALYSIS

### 2.3.4 PASS-BAND FILTERING

In terms of performance, filters can be classified as high-pass, low-pass, band-pass, all-pass, and reject-all filters. The frequency range that they allow to pass is known as the passband, and the frequency range that they do not allow to pass is known as the stopband. An ideal filter is one that completely rejects signals whose frequencies are not in the interval for which it was designed; unfortunately, such types of filter do not exist because of the physical limitations of the components with which they are manufactured. The response of an ideal filter can be approximated by mathematical functions, including Butterworth [4], Chebyshev [26], and elliptic filters [24]. In this manner, it can be stated that a band-pass filter is a melding of a low-pass filter and a high-pass filter. Therefore, we apply a band pass filtering processes, that is, from a RAW time series in $\varepsilon_{0-60}$, we generate two time series for the two bands whose behavior was observed to be the most relevant in $\Delta$. In other words, we created a filtered time series that contained only the band $\overline{\beta}$ or frequencies ranging from 12 to 21 Hz, and another filtered time series containing only the band $\overline{\gamma}$ or the frequency range between 30 and 45 Hz.

### 2.3.5 DEEP CONVOLUTIONAL NEURAL NETWORKS ARCHITECTURE

Figure 5(a) depicts the generated model in which a Deep Convolutional Neural Network (DCNN) is trained with four input signals: the image corpus $\Phi$ and $\pi$, along with the brain rhythms $\overline{\beta}$ and $\overline{\gamma}$. The objective is to obtain two classes at the output, namely, to predict whether a person likes or dislikes the functional product they consume. In addition, Figure 5(b) shows that the architecture of the DCNN model proposed here consists of multiple convolutional layers followed by batch normalization, max-pooling, and dropout layers to prevent overfitting. The architecture of the *ith* layer is defined as a repeated process with several convolutional layers, pooling and dropout layers. Equation 1 defines the general form of the first and subsequent convolutional layers, denoted as $\mathbf{H}_i$ where $i = 1...n = 3$, can be written as:

$$\begin{aligned} \mathbf{H}_i &= \text{ELU}(\mathbf{W}_i * \mathbf{D}_{i-1} + \mathbf{b}_i) & \mathbf{P}_i &= \text{MaxPool}(\mathbf{H}_i, \text{pool size} = 2 \times 2) \\ \mathbf{H}_i &= \text{BatchNorm}(\mathbf{H}_i) & \mathbf{D}_i &= \text{Dropout}(\mathbf{P}_i, \text{rate} = 0.5) \end{aligned} \tag{1}$$

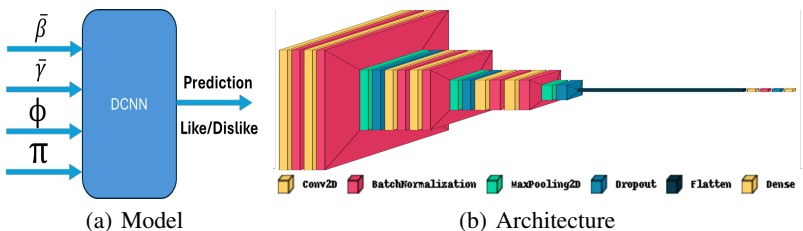

(a) Model           (b) Architecture

Figure 5: Deep Convolutional Neural Networks Architecture.

where $\mathbf{W}_i$ represents the filter weights $\mathbf{X}$ is the input and $\mathbf{b}_i$ is the bias term. The filter size is $5 \times 5$ with 64 units, $\mathbf{P}_i$ and $\mathbf{D}_i$ define MaxPool and Dropout layers, respectively. The Exponential Linear Unit (ELU) activation function is applied, and He normal initialization is used for the kernel. The output layer $\mathbf{O}$ is defined by Equation (2), which is a group of fully connected dense layers $\mathbf{F}_i$ with batch normalization using the Softmax activation function is defined as:

$$
\begin{aligned}
\mathbf{F}_i &= \mathrm{ELU}(\mathbf{W}f \cdot \mathrm{Flatten}(\mathbf{D}n) + \mathbf{b}_f) & \mathbf{O} &= \mathrm{Softmax}(\mathbf{W}_o \cdot \mathbf{F}_i + \mathbf{b}_o) \\
\mathbf{F}_i &= \mathrm{BatchNorm}(\mathbf{F}_i) & \mathrm{Optimizer} &= \mathrm{Adam}(\text{learning rate} = 0.001)
\end{aligned}
\tag{2}
$$

The model is compiled using the Adam optimizer with learning rate equal to 0.001 and the categorical cross-entropy loss function. Equations (1) and (2) detail the architecture illustrate the structured approach taken in our DCNN model to enhance the accuracy of feature extraction and classification using both PCA and DCNN methodologies. The proposed model, depicted by Figure 5(b), employs a sequential architecture optimized for image classification tasks. It begins with two convolutional layers, each with 64 filters of size $5 \times 5$, using the `ELU` activation function, `He-normal` initialization, and batch normalization, followed by max-pooling ($2 \times 2$) and dropout (40%) to prevent overfitting. This pattern is repeated with increased complexity in subsequent layers: two convolutional layers with 128 filters of size $3 \times 3$, and another block with 256 filters. Each convolutional block incorporates `ELU` activation, batch normalization, max-pooling, and progressively higher dropout rates, reaching 50%. The model then flattens the feature maps and adds a dense layer with 128 neurons, `ELU` activation, and a 60% dropout rate before the final `softmax` output layer for classification. It is compiled using the Adam optimizer with a learning rate of 0.001, categorical cross-entropy loss, and accuracy as the evaluation metric, ensuring robust learning and generalization.

## 3 RESULTS

### 3.1 EXPERIMENTS

To begin with, we present the results in two main scenarios when the customer's preference for a functional product is classified, that is, Like/Dislike classes: i) excluding $\overline{\beta}$ and $\overline{\gamma}$, or ii) including them to enhance the classification process, evaluating the effectiveness of a deep convolutional neural network model (DCNN) in categorizing customer preferences. Note that the image corpus was split into three categories, 70% for training, 20% for validation and finally 10% reserved for testing. To ensure reproducibility, this dataset used in this research is ready for download via this link. By publicly releasing this data set, other researchers will be able to reproduce our results, affirm our methodological process, and generate new paths of advancement for the field, a necessary component for transparency and collaboration in science. On the one hand, let us analyze the Receiver Operating Characteristics (ROC) curve, which depicts the true positive rate versus the false positive rate, demonstrating the balance between sensitivity and specificity at different threshold levels. Unfortunately, the ROC curve of the first version of our DCNN model did not show the expected sharp increase toward the upper left corner, as the ROC is equal to $0.73$, suggesting less than optimal performance to distinguish between classes.

Then, we evaluate the F1 score in Table 1, which combines precision and recall to provide a comprehensive assessment of the model's performance. The F1 score ranges from 0 to 1, with higher values indicating better accuracy. Unfortunately, our DCNN model produced an F1 score of $0.72$, which fell short of our expectations and revealed challenges in accurately classifying instances

into different categories. In general, the results of this first test using the DCNN model, without considering $\overline{\beta}$ and $\overline{\gamma}$, bring attention to the obstacles and constraints linked to its effectiveness in the first scenario. The problems faced emphasize the complexity of the classification assignment and stress the need for additional enhancement or adjustment.

<table>
<tr><td colspan="5">Table 1: Excluding $\overline{\beta}$ and $\overline{\gamma}$.</td></tr>
<tr><td></td><td>Precision</td><td>Recall</td><td>F1-score</td><td>Support</td></tr>
<tr><td>Dislike</td><td>0.82</td><td>0.68</td><td>0.74</td><td>40</td></tr>
<tr><td>Like</td><td>0.62</td><td>0.78</td><td>0.69</td><td>27</td></tr>
<tr><td>Accuracy</td><td></td><td></td><td>0.72</td><td>67</td></tr>
<tr><td>Macro avg</td><td>0.72</td><td>0.73</td><td>0.71</td><td>67</td></tr>
<tr><td>Weighted avg</td><td>0.74</td><td>0.72</td><td>0.72</td><td>67</td></tr>
</table>

<table>
<tr><td colspan="5">Table 2: Including $\overline{\beta}$ and $\overline{\gamma}$.</td></tr>
<tr><td></td><td>Precision</td><td>Recall</td><td>F1-score</td><td>Support</td></tr>
<tr><td>Dislike</td><td>0.94</td><td>0.99</td><td>0.96</td><td>77</td></tr>
<tr><td>Like</td><td>0.98</td><td>0.91</td><td>0.94</td><td>54</td></tr>
<tr><td>Accuracy</td><td></td><td></td><td>0.95</td><td>131</td></tr>
<tr><td>Macro avg</td><td>0.96</td><td>0.95</td><td>0.95</td><td>131</td></tr>
<tr><td>Weighted avg</td><td>0.96</td><td>0.95</td><td>0.95</td><td>131</td></tr>
</table>

Finally, we examine the confusion matrix, in Figure 6(b), which offers a glimpse into the effectiveness of the model by presenting the counts of true positive, true negative, false positive and false negative predictions. The analysis of this confusion matrix exposed a notable number of misclassifications in different categories, suggesting the existence of substantial classification inaccuracies. In this way, we present three highly encouraging ways to demonstrate the efficiency results of the **BRAIN** model, which includes only $\overline{\beta}$ and $\overline{\gamma}$ from the EEG time series: i) customer preference, ii) flavor of a sample and iii) flavor of a sample along with its customer preference. Performance metrics demonstrate exceptional accuracy and resilience, underscoring the efficacy of our approach in achieving research objectives.

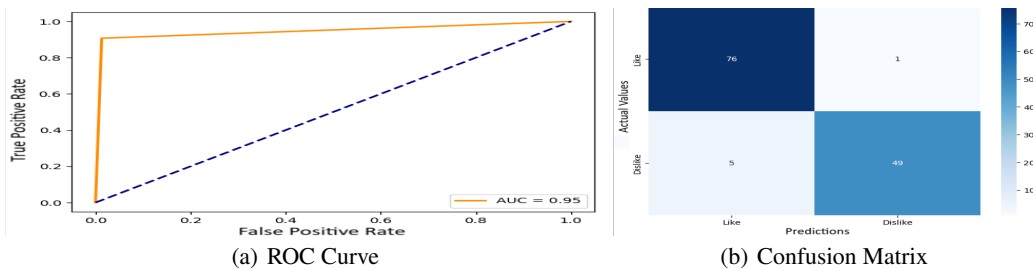

(a) ROC Curve                    (b) Confusion Matrix

Figure 6: Efficiency of **BRAIN** Architecture including $\overline{\beta}$ and $\overline{\gamma}$ brain rhythms in training, validation and test phases,when it is classified the customer's preference for a functional product.

Regarding customer preference, as shown in Table 2, our model demonstrated an exceptional F1 score of $0.95$, highlighting its impressive levels of precision and recall. The F1 score, which is a balanced measure of precision and recall, offers a thorough evaluation of the effectiveness of the classifier considering both false positives and false negatives. The achievement of such a high F1 score emphasizes the strength and dependability of our DCNN model in accurately categorizing data points in the dataset. Furthermore, as shown in Figure 6(a), the ROC curve, a key indicator of classifier performance, produced an impressive AUC of $0.95$. This means that our model exhibits remarkable discriminative capability, with a high true positive rate and a low false positive rate across a range of threshold values. Moreover, the confusion matrix produced by our model demonstrates outstanding performance in all categories, with each value exceeding $0.91$, as shown in Figure 6(b). This matrix visually illustrates the accuracy of the model's classification by presenting the proportions of true positive, true negative, false positive, and false negative predictions. The consistently high metrics in the confusion matrix confirm the model's ability to classify instances accurately in all categories while keeping misclassification errors to a minimum. In general, the results of this first test using the DCNN model, without considering $\overline{\beta}$ and $\overline{\gamma}$, bring attention to the obstacles and constraints linked to its effectiveness in our particular scenario. The problems faced emphasize the complexity of the classification assignment and stress the need for additional enhancement or adjustment.

From Figure 7 and , when classified as the flavor of a functional product that a customer perceives, namely acidic, bitter, salty or sweet, our DCNN model maintains outstanding performance in different evaluation metrics taking into account both $\overline{\beta}$ and $\overline{\gamma}$. This includes an F1 score of $0.96$ and a confusion matrix with values greater of $0.91$ for all components. These results signify the efficiency of our

Table 3: Efficiency of **BRAIN** Architecture: classifying the flavor of a functional product.

|  | Precision | Recall | F1-score | Support |
|---|---|---|---|---|
| Acidic | 0.91 | 1.00 | 0.96 | 96 |
| Bitter | 1.00 | 1.00 | 1.00 | 96 |
| Salty | 1.00 | 0.84 | 0.92 | 96 |
| Sweet | 0.94 | 1.00 | 0.97 | 96 |
| **Accuracy** |  |  | 0.96 | 384 |
| **Macro avg** | 0.96 | 0.96 | 0.96 | 384 |
| **Weighted avg** | 0.96 | 0.96 | 0.96 | 384 |

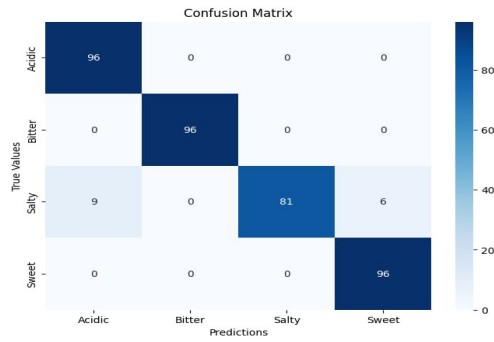

Figure 7: Confusion Matrix of **BRAIN** Architecture.

approach and its suitability for application in various domains that demand accurate and reliable classification, Figure 7.

Table 4: Efficiency of **BRAIN** Architecture: classifying flavor and preference.

|  | Precision | Recall | F1-score | Support |
|---|---|---|---|---|
| Acidic Dislike | 0.81 | 0.85 | 0.83 | 26 |
| Acidic Like | 0.94 | 0.92 | 0.93 | 66 |
| Bitter Dislike | 1.00 | 0.92 | 0.96 | 65 |
| Bitter Like | 0.86 | 1.00 | 0.92 | 30 |
| Salty Dislike | 0.79 | 0.93 | 0.85 | 44 |
| Salty Like | 0.92 | 0.75 | 0.83 | 48 |
| Sweet Dislike | 0.82 | 0.75 | 0.78 | 12 |
| Sweet Like | 0.95 | 0.98 | 0.96 | 83 |
| **Accuracy** |  |  | 0.91 | 374 |
| **Macro avg** | 0.89 | 0.89 | 0.88 | 374 |
| **Weighted avg** | 0.91 | 0.91 | 0.91 | 374 |

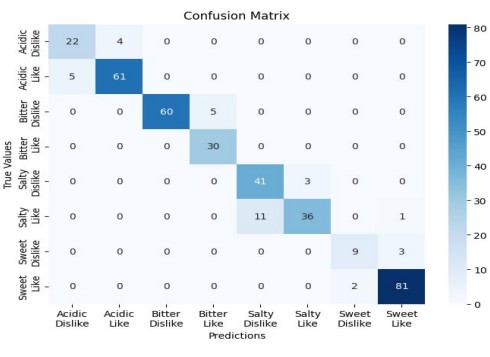

Figure 8: Confusion Matrix of **BRAIN** Architecture: classifying flavor and preference.

Finally, from Table 4, when classified not only the flavor, but also the customer's preference for a functional product, namely whether acidic, bitter, salty, or sweet flavor, the customer likes it or not, this gives as a result of the eighth instance. In this scenario, our DCNN model has demonstrated exceptional performance in various evaluation metrics, incorporating both $\overline{\beta}$ and $\overline{\gamma}$. These metrics include an F1-score average of $0.91$, and a confusion matrix with values greater than $0.78$ for all components. These findings substantiate the robustness of our methodology and its suitability for deployment in multiple domains that require precise and reliable classification, Figure 8.

### 3.2 COMPARISON OF PROPOSED ARCHITECTURE VS WELL-KNOWN MODELS

The same data set was used to test different architectures in terms of the image classification measure of the proposal, so VGG16, EfficientNetB0, and ResNet50 are tested with the same input and classification objective. All three CNN architectures followed the same approach by treating models pre-trained on ImageNet as feature extractors and freezing their layers to preset them with prior knowledge. All models are fine-tuned for a custom binary classification task (i.e. *Like* and *Dislike*) with RGB input images of size $224 \times 224$ pixels. The custom classification head added on top of all three architectures is a Flatten layer followed by Dense with 256 units and `ReLU` activation, Dropout with rate 50% to avoid overfitting, and finally Dense layer with `softmax` activation for classification. A learning rate of $0.0001$, categorical cross-entropy loss, and accuracy evaluation metric are used to compile the three models, using `Adam` optimizer. While these backbone architectures are similar in terms of core structure and fine-tuning methodology, they differ: VGG16 is simple and deep with its convolutional layers; EfficientNetB0 uses an efficient scaling method; and ResNet50 relies on residual connections to optimize learning and performance. These settings use a pre-trained feature extraction leading to task-specific minimal customization yielding accurate and efficient classification

performance. It is important to mention that using identical image database, EfficientNetB0 and ResNet50 architectures produce an AUC-ROC of 0.57 and an F1-Score of 0.59 as reported; the results are shown respectively in Figures 9(a) and 9(b). 77 images are actually classified as *Like* and 54 of them are incorrectly processed as *Like* also (Among 131 test cases) In addition, models never predict any of the cases as *Dislike*. In comparison, a VGG16 architecture performs markedly better with AUC-ROC achieving 0.91 and F1-Score 0.82 Since the VGG16 gives results above 90%, we need to dive deep into this architecture, Figure 9(c). VGG16 obtains a confusion matrix that shows how well VGG16 model predicts whether someone *Likes* or *Dislikes* something. The model got 65 *Like* predictions and 42 *Dislike* predictions correct. However, it made mistakes on 12 occasions for each category—mislabeling 12 *Dislikes* as *Likes* and 12 *Likes* as *Dislikes*.

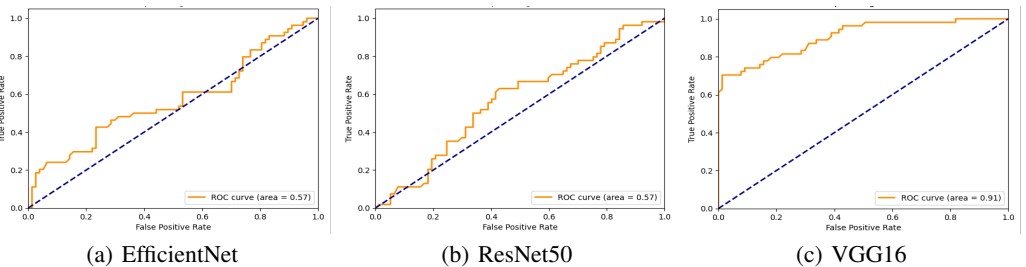

| (a) EfficientNet | (b) ResNet50 | (c) VGG16 |
|---|---|---|

Figure 9: AUC-ROC Curve: Analysis of (a) EfficientNet, (b) ResNet50 , and (c) VGG16 Models.

### 3.3 EMOTIONAL REACTION DURING THE EXPERIMENT

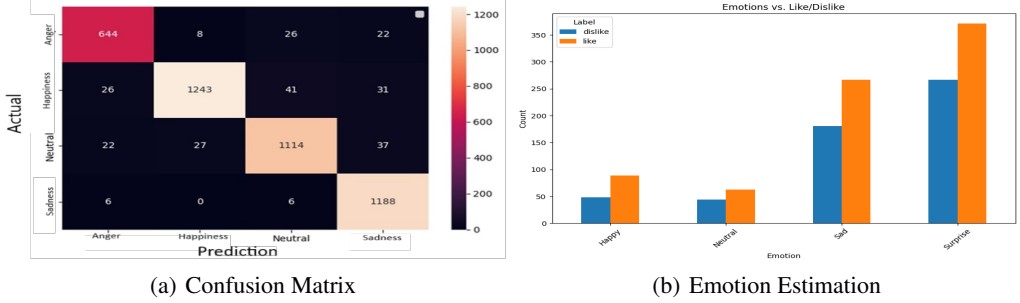

| (a) Confusion Matrix | (b) Emotion Estimation |
|---|---|

Figure 10: Emotional Reaction during the experiment: (a) Confusion Matrix, and (b) Emotion Estimation during the experiment.

With functional products, we did not measure the emotion where associated with the during tasting experience. Thus, we trained the same architecture shown in Figure 5(b) with a corpus of images which can be downloaded via this link. The DCNN was trained with a corpus of 30,948 images for training, 8,899 images for validation and 4441 images for testing in order to classify four feelings: happy, sad, angry and neutral. After training and validation stages, this model obtains 94% in terms of F1-score, indicating that the model can make accurate and reliable predictions with previously unseen data, achieving effective detection of the emotion of a human being. This model achieves good performance not only with overall accuracy, but also has the confusion matrix detailing in Figure 10(a) where it correctly classifies all emotions at a rate of 92% to 96%.

In this way, using transfer learning to this proposal; the emotional experiences recorded were quite different among categories when consuming functional products, Figure 10(b). The happy, neutral, sad and surprise were connected to the positive express emotion: *Like* with 89,63,267, and 371 respectively. On the other hand, negative emotions such as *Dislike* were less common (sadness: 181; surprise: 267), and happiness and neutrality even rarer (48; 44). The emotional patterns correspond to different sensory experiences with functional products. Specifically, when the product exceeds expectations on attributes such as taste, texture or perceived health benefits they can elicit positive surprise and happiness. In the same vein, sadness may be indicative of an implicit link between functional products and health-related aspects or life-style changes their manufacturers frequently associate with such products. The results highlight the complex interplay of emotions in shaping consumer perceptions and emphasize the importance of addressing both sensory and psychological factors in product development.

## 3.4 DISCUSSION

Principal component analysis (PCA) as a preprocessing step before training Deep Convolutional Neural Networks (DCNN) addresses several challenges. PCA reduces data dimensionality while preserving essential variance, crucial for high-dimensional EEG signal processing [12]. It extracts significant neuronal metrics (for example, low beta $\overline{\beta}$ and gamma $\overline{\gamma}$ frequency bands, attention and meditation levels) from EEG datasets [23]. The application of PCA before DCNN, compared to raw EEG/facial data. PCA reduces overfitting in DCNN, improves training by removing noise and irrelevant features, and improves performance in decoding consumer behavior [21]. Incorporating PCA before DCNN simplifies high-dimensional data, increasing computational efficiency and reducing resource utilization without compromising predictive power or accuracy [18]. In addition, it improves the interpretation of EEG signals, crucial in neuromarketing to uncover the neural mechanisms behind consumer decision making [15]. The reason for using PCA as the main approach is that it is the most widely used method among neuroscience studies when it comes to dimensionality reduction and feature extraction. Principal component analysis (PCA) allows the discovery of important components that contain most of the variance in the data, which is also useful because EEG signals are highly complex. Although deeper machine learning approaches, for example, t-SNE, autoencoders, could provide additional insights, setting a strong baseline is needed and have interpretable results. Future versions of this study may integrate more advanced algorithms to investigate the possibility of improved classification ability and insights. In conclusion, the combination of PCA and DCNN uses the strengths of both techniques, ensuring a powerful and interpretable model. This approach improves robustness and applicability, aligning with the development of precise and reliable classification systems, particularly in personalized nutrition strategies for the food sector. With these compelling results, we firmly establish that **BRAIN** has the unparalleled ability not only to categorize the nuanced flavors of the functional product, but also to discern the personal preferences of individuals with remarkable precision, achieving a classification error less than 10%. This milestone unequivocally underscores the robustness and precision of our system, making an indelible mark on the future of personalized consumer experiences.

## 4 CONCLUSIONS

This study illustrates how brain activity can identify participants' food preferences. EEG analysis indicated that EEG effectively assesses taste preferences. By analyzing EEG data, we identified whether participants liked or disliked particular food samples. The key decision-making factors were low beta and low gamma frequency bands, along with Attention and Meditation levels. These data are crucial for the food industry, providing valuable insights for companies aiming to enhance the attractiveness of functional foods. Analyzing brain signals in response to tastes allows researchers to identify preferred flavors for individuals. The Low Beta ($\overline{\beta}$) and Low Gamma ($\overline{\gamma}$) bands are significant in the decision-making process. The results of the initial DCNN model test, excluding $\overline{\beta}$ and $\overline{\gamma}$, highlight challenges and limitations. The difficulties encountered underscore the complexity of the categorization task, necessitating further refinement. Ultimately, our DCNN model exhibited remarkable performance with $\overline{\beta}$ and $\overline{\gamma}$ included. This includes a ROC curve with an AUC of $0.95$, an F1 score of $0.95$, and a confusion matrix with values exceeding $0.91$. These findings indicate our approach's effectiveness and suitability for precise and reliable classification across various fields. This study gives some interesting insights into consumer neuroscience and neuromarketing, but there are also limitations. The sample size of 16 participants aged 20 to 29 may not represent diverse consumer preferences. The proposed architecture itself needed to be assessed for efficiency given the relatively small sample size. It was thus compared against the baseline performance of popular models like EfficientNet, ResNet and VGG16. The proposed architecture outperformed these models by at least 5% as shown in the demonstration. Some good here use of transfer learning suited for functional products (category specific emotional experiences) Positive emotions such as Happiness, Neutrality, Sadness and Surprise were associated with *Like*, whereas Negative emotions such as *Dislike* occurred less frequently. These findings highlight the complex emotional drivers in consumer perception and suggest that both sensory attributes and psychological factors should be taken into account for product design. Future work includes the integration of additional mostly biosignals, such as electrocardiograms in elaborate programming architectures utilizing distributed or parallel computing. It opens the door to integrating diverse biosignals and high-resolution qEEG imaging, a major advancement in the processing and analysis of these biosignals.

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
