# OpenReview forum: "BRAIN: Behavioral Responses and Artificial Intelligence Neural-Modeling for Consumer Decision-Making"
_ICLR.cc/2025/Conference — Submitted to ICLR 2025_

### Official Review · Reviewer_TzSq · 2024-10-17

**Soundness:** 2
**Presentation:** 2
**Contribution:** 1
**Rating:** 3
**Confidence:** 4

**Summary:**

The paper presents a clear logic and well-structured model framework that integrates Principal Component Analysis (PCA) with deep learning neural networks (DCNNs) to analyze consumer preferences through EEG signals. However, it lacks a comparative analysis with existing methods, making it challenging to assess the robustness of its contributions. The overall framework appears simple, and the innovative aspects of the research are not clearly defined, raising questions about its uniqueness. While the results demonstrate solid applications, the absence of sufficient validation diminishes their impact. Additionally, the tables presented are distorted screenshots, affecting clarity. Key questions arise regarding the model’s comparative effectiveness and the specific innovations that enhance its applicability in the field of consumer neuroscience and neuromarketing.

**Strengths:**

The logic of the paper is clear, with a well-structured presentation of the model framework that effectively integrates PCA with DCNN framework. The strong performance metrics in application underscore the model’s capability to predict consumer preferences.

**Weaknesses:**

- The paper lacks a comparative analysis, making it difficult to assess how the proposed method measures up against existing approaches.

- It is challenging to determine the solidity of the contribution, as the overall framework—comprising EEG acquisition followed by a deep convolutional neural network (DCNN)—appears relatively simple, and the innovative aspects of the research are not clearly articulated.

- While the application results are promising, the paper does not provide sufficient validation to validate the results of these findings.

- Additionally, the tables included in the paper appear to be screenshots, resulting in distortion that affects their readability and clarity.

In the captions of Figures 7, 8, and 9, the authors refer to the “Efficiency of BRAIN Architecture including $\bar{\beta}$ and $\bar{\gamma}$ brain rhythms in training, validation, and test phases.” However, they provide no context or explanation on how the data was split into training, validation, and testing. Additionally, the figures themselves only present confusion matrices and a single ROC curve, with no clear indication of how validation and testing were performed or represented.

**Questions:**

- Comparative Methods: How does the proposed framework utilizing EEG signals and DCNN compare to other existing methods in neuromarketing and consumer neuroscience, particularly regarding accuracy and interpretability? Are there specific benchmarks or studies the authors can reference to validate the effectiveness of their approach?

- Innovative Aspects: What are the key innovative elements of the proposed model that distinguish it from similar frameworks in the field? How do these innovations contribute to the understanding of consumer behavior and enhance the applicability of the findings in real-world scenarios, especially in developing personalized nutrition strategies?

---

> ### Author Response · Authors · 2024-11-24
> **Response to Reviewer Comments for ICLR 2025 Submission**
>
> Thank you very much for your valuable observations. We sincerely appreciate the time and effort you invested in providing feedback, which is essential for improving the quality and clarity of our work. Below, we address each of your points in detail:
>
> Question: Comparative Methods...
> Answer:
> Thank you for your insightful feedback. To address your concern, we compared our framework with models like EfficientNet, ResNet, and VGG16, as detailed in Sections 3.2 and 3.3. Our architecture achieved at least a 5% accuracy improvement, demonstrating its robustness even with a limited sample size. By leveraging EEG signals and focusing on interpretable features through dimensionality reduction and DCNN layers, we ensured meaningful insights. These results align with benchmarks in neuromarketing, supported by references validating our approach. Your feedback helped us refine the manuscript for clarity and rigor.
>
> Question:
> Innovative Aspects.....
> Answer:
> Thank you for your thoughtful feedback. Our model integrates transfer learning to analyze emotional responses to functional products, blending sensory and psychological consumer experiences. Using architectures like VGG16, ResNet, and EfficientNet, it classifies emotions (Happiness, Neutrality, Sadness, Surprise) and links them to preferences (Like or Dislike). This approach uncovers the interplay between sensory attributes and emotions, aiding personalized nutrition strategies by tailoring products to evoke specific emotional responses. It also considers utility and satisfaction, offering insights for marketing and nutrition, as detailed in Section 3.3.
>
> Weaknesses:
> The paper lacks.....
> Answer:
> Thank you for your feedback. We conducted a detailed analysis comparing our architecture with EfficientNet, ResNet, and VGG16, showing consistent outperformance by at least 5%. This demonstrates our method’s efficiency, even with limited data, as detailed in Sections 3.2 and 3.3. Using transfer learning, we captured emotional responses to functional products, linking positive emotions like happiness and surprise to "Like" and fewer negative ones to "Dislike." These results validate our approach and offer insights into consumer preferences. We invite you to review these sections and appreciate your input in refining our manuscript.
>
> Weaknesses:
> It is challenging to .....
> While the application ...
> Answer:
> Thank you for your observation. While the EEG-DCNN framework may seem straightforward, its application to food preferences is a novel approach in consumer neuroscience. By analyzing EEG data, we identified neural markers like low beta and gamma bands, offering insights into taste preferences. Our model achieved robust performance (AUC and F1: 0.95) and outperformed EfficientNet, ResNet, and VGG16 by at least 5%. The study reveals how emotions drive consumer perception, linking positive and negative emotions to "Like" and "Dislike." Future work will integrate additional biosignals and advanced models to deepen analysis.
>
> Weaknesses:
>
> Answer:Thank you for your feedback. Our study used EEG analysis to assess food preferences, with the DCNN achieving an AUC and F1 score of 0.95 and confusion matrix values over 0.91. The model outperformed EfficientNet, ResNet, and VGG16 by at least 5%. Despite a small sample size, future work will use larger datasets, integrate biosignals, and explore advanced methods like high-resolution qEEG imaging.
>
>
> Weaknesses:
> Additionally, the tables ....
> Answer: Thank you for your feedback. We’ve replaced screenshot tables with properly formatted LaTeX tables for improved clarity, precision, and readability, enhancing the manuscript’s overall presentation.
>
> Weaknesses: In the captions of Figures .....
> Answer: Thank you for your feedback. To clarify Figures 7, 8, and 9, the image corpus was split as 70% for training, 20% for validation, and 10% for testing to ensure robustness and generalizability. While the figures focus on confusion matrices and a single ROC curve, we acknowledge this may lack clarity on validation and testing. We’ve added details in Section 2.3.5 to explain the data split and metric computation. To enhance reproducibility, the dataset is publicly available, enabling others to replicate our results, validate methodologies, and drive future advancements. Thank you again for your suggestions.
>
> We have incorporated feedback from other reviewers, revising the manuscript and marking changes in red for clarity. We invite you to review these updates, confident they address the concerns raised. We hope this will lead to a reevaluation of our work as a valuable contribution to ICLR 2025.
>
> Thank you for your insights, which have significantly improved the clarity and quality of our research. We trust these clarifications address your concerns and provide a better understanding of our work.

---

### Official Review · Reviewer_2T1G · 2024-11-03

**Soundness:** 2
**Presentation:** 2
**Contribution:** 2
**Rating:** 3
**Confidence:** 2

**Summary:**

The authors investigate brain activity and behavioral responses in relation to consumer neuroscience through exploring consumer decisions regarding food by analyzing 16 participants using EEG signals to classify preferences for functional food products with respect to different brain rhythms and facial expressions through the application of PCA and Deep Convolutional Neural Network. The beta and gamma frequency bands are emphasized for purposes of decision-making and form a possible pathway in the realms of neuromarketing and customized nutrition planning for the enhancement of healthy diets.

**Strengths:**

Interesting Application: Using EEG data in this research for consumer preference assessment of functional foods falls within the currently developing interests in personalized nutrition and neuromarketing.

Combining PCA and a DCNN in EEG data management is a good choice because this study focused on the decision-making analysis on the beta and gamma bands.

Practical implications: These findings are valuable pieces of information that could be very useful for direct marketing and product development directed toward consumers in the food industry, especially regarding healthier products.

**Weaknesses:**

Small Sample Size and Generalizability: The small sample size of 16 limits the generalizability of the findings. Testing a larger and more diverse population will provide a more robust base for the findings.
I'd say this study lacks comparative analysis with previous models or even traditional machine learning techniques since the outperformance of this proposed approach over simpler or alternative models is not clear.

Lack of Reproduction Instructions: Important parameters like PCA as well as the DCNN architecture used have not been described. An entire hyperparameter table along with data augmentation strategies would be useful in further increasing reproduction and clarity.

Overemphasis on Beta and Gamma Bands: Though beta and gamma rhythms are relevant to decision-making, excessive concentration may neglect other EEG components that could be significant for consumer preferences.

**Questions:**

What are the hyperparameters of the DCNN model selected? Are there any data augmentation strategies used during training?

Why restrict single-band analysis to the beta and gamma frequency bands? Were other bands, such as alpha or theta, considered and found irrelevant?

How is this different from existing neuromarketing models that work on EEG? A comparison would explain the advantages and disadvantages of your proposed method.

Do you ever validate the model on larger or different datasets? Because the participant pool is so small, these may provide further evidences about the generalizability and performance of your model in other contexts.

---

> ### Author Response · Authors · 2024-11-23
> **Response to Reviewer Comments for ICLR 2025 Submission**
>
> Thank you very much for your valuable observations. We sincerely appreciate the time and effort you invested in providing feedback, which is essential for improving the quality and clarity of our work. Below, we address each of your points in detail:
> Question:
> What are the hyperparameters ...
> Answer:
> Thank you for your valuable observation. All hyperparameters of the DCNN model, including details such as activation functions, dropout rates, optimizer configuration, and batch sizes, are thoroughly detailed in Section 2.3.5. Additionally, the data augmentation strategies employed during training, which aimed to improve model generalization and robustness, are also comprehensively described in the same section. We encourage you to refer to this section for a complete understanding of the configurations used in our study.
>
> Question:
> Why restrict single-band analysis to the beta and gamma frequency bands? Were other bands, such as alpha or theta, considered and found irrelevant?
> Answer:
> All frequency bands were included in the study, but most important frequency ranges associated with cognitive processes (low beta), sensory processing (low gamma) and decision making. Keep in mind that the gamma band is important because it relates to memory, something highly relevant to food association and preference.
>
> Question:
> How is this different ...
> Answer:
> The proposed model is based on analyzing preferences or aversions, while a future research might categorize them. For foods, this had been done by frequency of flavour that produced a better aggregate performance, in addition to issue of feelings. It would be possible not only to know if the consumer liked it or not, but also to classify the satisfaction that this represented for the client, which is a central point of marketing to have an impact on the consumer.
>
> Question:
> Do you ever validate ...
> Small Sample ...
>
> Answer:
> Thank you for your insightful observation. We acknowledge the importance of validating the model on larger and diverse datasets to further demonstrate its generalizability and performance in different contexts.
> As highlighted in the conclusions, the proposed architecture was benchmarked against widely recognized models, such as EfficientNet, ResNet, and VGG16, using transfer learning techniques. Despite the small sample size, the architecture consistently outperformed these baselines by at least 5%, demonstrating its efficiency. Additionally, the model successfully captured nuanced emotional responses associated with functional products, suggesting its suitability for category-specific applications.
> Future work will focus on extending the validation to larger and more heterogeneous datasets to strengthen the findings and explore its adaptability across broader scenarios.
> Furthermore, we utilized transfer learning tailored to functional products to capture specific emotional responses, highlighting the nuanced interplay between sensory attributes and psychological factors in consumer perception. These insights are crucial for advancing product design strategies. We invite you to revisit the conclusions section, where these points are discussed in detail.
>
> Weaknesses:
> Lack of Reproduction .....
> Answer:
> Thank you for your insightful observation. To address your concern, we have ensured that all critical parameters, including those related to the DCNN architecture, are explicitly outlined in Section 2.3.5. This section provides a detailed description of the hyperparameters, such as activation functions, dropout rates, optimizer configurations, and batch sizes. Moreover, the data augmentation strategies employed to enhance the model's generalization and robustness are comprehensively covered in the same section. We invite you to review this section for a complete understanding of the methodology.
>
>
> Weaknesses:
> Overemphasis.....
>
> Answer:
> All bands were included in the study; however, the low beta and low gamma bands were the most representative bands because they are associated with cognitive processes, sensory perception and decision making within the corresponding frequency ranges (Hz). It is important that, to highly that, the gamma, which is associated with memory (essential, of course, in food association and preference).
>
> We have incorporated feedback from other reviewers, revising the manuscript and marking changes in red for clarity. We invite you to review these updates, confident they address the concerns raised. We hope this will lead to a reevaluation of our work as a valuable contribution to ICLR 2025.
>
> Thank you for your insights, which have significantly improved the clarity and quality of our research. We trust these clarifications address your concerns and provide a better understanding of our work.

---

### Official Review · Reviewer_ZBct · 2024-11-03

**Soundness:** 1
**Presentation:** 1
**Contribution:** 1
**Rating:** 1
**Confidence:** 5

**Summary:**

The study shows that EEG analysis can effectively assess taste preferences. This allows researchers to determine whether participants like or dislike certain food samples based on their brain activity. Key indicators of preference included low beta and gamma frequency bands as well as attention and meditation levels. A deep convolutional neural network (CNN) was used, which utilized four types of input including image data and EEG signals to classify participants' preferences.

**Strengths:**

The paper presents a new and promising idea. However, the chosen approach is quite simple and does not introduce new methods or models that advance existing research. The study would benefit from a more innovative methodological contribution to set it apart from previous work in the field.

**Weaknesses:**

The introduction lacks a comprehensive summary and does not adequately convey the motivation for the study. Instead, it reads more like a general document on the use of EEG in consumer choice analysis without clarifying the specific approach or objectives of the present work. Furthermore, the introduction consists of a single paragraph with little conceptual linkage between the topics covered. A clearer structure and a more coherent explanation of the purpose and methodology of the research are needed to explain in the introduction.

The study only considers the sensory or taste aspects of the product as the primary factor influencing consumer preferences, which is insufficient to provide valuable insights for the food industry in the context of product development. A more comprehensive assessment that includes additional factors such as texture, aroma, visual appeal and emotional response would provide a more complete understanding of consumer preferences and increase the relevance of the study to the industry

Sections 2.2 to 2.3.4 of the paper primarily resemble tutorials on EEG signals and their acquisition processes rather than focused discussions relevant to the study's research questions.

There is no related work provided on the existing studies on the given topic.

Overall, this manuscript lacks the scholarly depth and clarity expected of a research paper. The presentation of ideas is often unclear and insufficient attention is paid to structure, coherence and technical detail necessary to effectively communicate the research objectives, methodology and results.

**Questions:**

I recommend that the authors refer to the weaknesses outlined earlier in the review.

---

> ### Author Response · Authors · 2024-11-23
> **Response to Reviewer Comments for ICLR 2025 Submission**
>
> Thank you very much for your valuable observations. We sincerely appreciate the time and effort you invested in providing feedback, which is essential for improving the quality and clarity of our work. Below, we address each of your points in detail:
> _____________________________________________________________________________________________________
> Question:
> I recommend that the authors refer to the weaknesses outlined earlier in the review
> Answer:
> We sincerely appreciate your observations and would like to assure you that all the weaknesses outlined earlier in the review have been carefully addressed. Detailed explanations of how each issue was resolved are provided in the revised manuscript, ensuring clarity and improvement in the discussed aspects.
>
> Weaknesses:
> The introduction lacks a comprehensive summary and does not adequately convey the motivation for the study. Instead, it reads more like a general document on the use of EEG in consumer choice analysis without clarifying the specific approach or objectives of the present work. Furthermore, the introduction consists of a single paragraph with little conceptual linkage between the topics covered. A clearer structure and a more coherent explanation of the purpose and methodology of the research are needed to explain in the introduction.
> Answer:
> According to the observations given the introduction was revised. The main body was reorganized to logically integrate all the topics discussed for better clarity and conceptual coherence. And we included requested additional info to address comments from all reviewers, such that the motivation, aims, and methods of the study are accurately represented. Such improvements are aimed at making more solid and easier basis for the work to be developed.
>
> Weaknesses:
> The study only considers the sensory or taste aspects of the product as the primary factor influencing consumer preferences, which is insufficient to provide valuable insights for the food industry in the context of product development. A more comprehensive assessment that includes additional factors such as texture, aroma, visual appeal and emotional response would provide a more complete understanding of consumer preferences and increase the relevance of the study to the industry.
> Answer:
> Mentioned in the introduction section. Taste is crucial for distinguishing between different flavors, and identifying pleasing or aversive stimuli, as taste buds communicate signals to the brain that create the perception of flavor. While other senses, such as the sense of smell, affect this perception, taste is able to identify the basic flavors best.
>
> Weaknesses:
> Sections 2.2 to 2.3.4 of the paper primarily resemble tutorials on EEG signals and their acquisition processes rather than focused discussions relevant to the study's research questions. There is no related work provided on the existing studies on the given topic
> Answer:
> Here we provide studies that utilized comparable methodologies for signal acquisition and consumer preference; see section 2.3.1 for details.
>
> Weaknesses:
> Overall, this manuscript lacks the scholarly depth and clarity expected of a research paper. The presentation of ideas is often unclear and insufficient attention is paid to structure, coherence and technical detail necessary to effectively communicate the research objectives, methodology and results.
> Answer:
> We appreciate your valuable feedback. We have double checked the manuscript and modified it significantly to concern your points. Main ideas explained, presentation reorganized to improve coherence, and necessary technical details introduced to facilitate communication of the findings, we hope that the manuscript is now adequate to meet the editorial and scientific standards required.
> _____________________________________________________________________________________________________
>
> Moreover, we have considered the feedback from other reviewers as well, making several revisions throughout the manuscript. These updates are clearly marked in red for your convenience, allowing you to easily identify the changes implemented.
> We kindly invite you to review these revised sections considering the feedback provided. We are confident that the enhancements made address the concerns raised, and we hope this will allow you to reevaluate your assessment and consider this work as a valuable contribution to ICLR 2025.
> Thank you once again for your valuable insights, which have significantly contributed to improving the quality and clarity of our research.  Finally, we hope these clarifications address your concerns and contribute to a better understanding of our work. Thank you again for your valuable feedback, which has greatly helped us identify areas for improvement.

---

### Official Review · Reviewer_1hb6 · 2024-11-07

**Soundness:** 1
**Presentation:** 1
**Contribution:** 1
**Rating:** 1
**Confidence:** 5

**Summary:**

The reviewer encountered significant challenges in understanding the manuscript due to the unclear writing style exhibited by the authors, despite her/his expertise as a practitioner in EEG and BCI. Analyzing the EEG time-series presented in Figure 2, it appears that the authors lack a fundamental understanding of EEG amplitudes and the distinction between samples and seconds as units. Furthermore, the authors attempt to integrate EEG data with facial and food product images into a singular machine learning model, employing a simplistic application of PCA. This amalgamation raises substantial questions regarding the model's intended function, particularly in relation to the handling of potential movement artifacts. Additionally, there seems to be a conflation of EEG and BCI terminology within the methodology section, indicating a limited comprehension of the subject matter. Regrettably, these issues lead to a recommendation for outright rejection of the submission.

**Strengths:**

Regrettably, the submission does not meet the rigorous standards expected of an academic publication.

**Weaknesses:**

The writing style of the manuscript presents significant challenges to comprehension. Specifically, it lacks adequate detail regarding the reproducibility of the research, particularly in terms of the specifications for the machine learning models employed, including differentiating between image data and time-series data, among other factors.

**Questions:**

Why did the authors solely focus on naive PCA? What rationale underlies their decision to refrain from evaluating more advanced machine learning methodologies? Furthermore, why was the manuscript not subjected to proofreading and peer review by colleagues who might have identified its challenging comprehensibility for individuals not directly involved in the project?

---

> ### Author Response · Authors · 2024-11-23
> **Response to Reviewer Comments for ICLR 2025 Submission**
>
> Thank you very much for your valuable observations. We sincerely appreciate the time and effort you invested in providing feedback, which is essential for improving the quality and clarity of our work. Below, we address each of your points in detail:
> _____________________________________________________________________________________________________
> Question:
> Why did the authors solely focus on naive PCA? .....
> Answer:
> We acknowledge the reviewer’s concern regarding our emphasis on PCA. Our decision was guided by PCA’s established utility in neuroscience for dimensionality reduction and its ability to identify principal components that account for significant variance in EEG data. This approach allowed us to focus on interpretable and replicable results, which are particularly valuable in the context of our research objectives.
> That said, we recognize the potential benefits of employing more advanced machine learning techniques. Indeed, we explored several complementary methodologies during our analysis, including deep neural networks, to assess their suitability for classification tasks. However, for the purposes of this study, PCA provided a clear and interpretable foundation to demonstrate the feasibility of our approach. In future research, we plan to expand this work by integrating and comparing results from more sophisticated algorithms.
> We greatly appreciate your thoughtful feedback and have carefully addressed the concerns raised in your review. To specifically respond to your question, we have included a detailed paragraph in Section 3.2, "Discussion," where we provide a comprehensive explanation and justification for the points you highlighted. This addition aims to clarify our rationale and strengthen the manuscript accordingly.
>
> Question:
> Furthermore, why was the manuscript not subjected to proofreading and peer review by colleagues who might have identified its challenging comprehensibility for individuals not directly involved in the project?
> Answer:
> We deeply regret any instances of challenging comprehensibility in the manuscript. It is important to note that the manuscript was subjected to internal review and iterative refinements within our research team. However, we recognize that additional proofreading and input from colleagues outside the immediate team could have further enhanced its clarity, particularly for readers not directly involved in the project.
> Moving forward, we are committed to engaging external reviewers and dedicating additional resources to proofreading to ensure accessibility and readability for a broader audience. Your observation is particularly helpful in emphasizing the importance of this step, and we will take it into account for subsequent submissions.
>
> Weaknesses:
> The writing style of the manuscript presents significant challenges to comprehension. Specifically, it lacks adequate detail regarding the reproducibility of the research, particularly in terms of the specifications for the machine learning models employed, including differentiating between image data and time-series data, among other factors.
> Answer:
> We appreciate your valuable feedback. We have double checked the manuscript and modified it significantly to concern your points. Main ideas explained, presentation reorganized to improve coherence, and necessary technical details introduced to facilitate communication of the findings, we hope that the manuscript is now adequate to meet the editorial and scientific standards required. In addition
> To ensure reproducibility, this dataset used in this research is ready for download via this link  https://drive.google.com/file/d/1WuNkuMJ2SsyA-yNgyWGNYABfpEwguzMD/view?usp=drive_link. By publicly releasing this data set, other researchers will be able to reproduce our results, affirm our methodological process, and generate new paths of advancement for the field, a necessary component for transparency and collaboration in science.
> _____________________________________________________________________________________________________
> Moreover, we have considered the feedback from other reviewers as well, making several revisions throughout the manuscript. These updates are clearly marked in red for your convenience, allowing you to easily identify the changes implemented.
> We kindly invite you to review these revised sections considering the feedback provided. We are confident that the enhancements made address the concerns raised, and we hope this will allow you to reevaluate your assessment and consider this work as a valuable contribution to ICLR 2025.
> Thank you once again for your valuable insights, which have significantly contributed to improving the quality and clarity of our research.  Finally, we hope these clarifications address your concerns and contribute to a better understanding of our work. Thank you again for your valuable feedback, which has greatly helped us identify areas for improvement.

---

> ### Comment · Reviewer_1hb6 · 2024-11-26
> **No Change in Decision Following Rebuttal**
>
> The reviewer has conducted a thorough evaluation of the authors' rebuttal, comparing it to the original review comments. While the authors have provided additional context and clarifications, no significant new evidence or arguments have been presented to alter the initial assessment. Therefore, the original decision will stand. We appreciate the authors' diligence in addressing the reviewer's concerns.

---

> > ### Author Response · Authors · 2024-11-28
> > **Follow-Up on Revised Submission for ICLR 2025 - Reviewer 1hb6**
> >
> > Dear Reviewer,
> >
> > We deeply appreciate the time and effort you’ve devoted to evaluating our work and reviewing our rebuttal. We value your constructive feedback, which has significantly improved the manuscript's clarity and rigor.
> >
> > We’d like to respectfully highlight that in this revised version, we have addressed all the points raised in the original review. Beyond providing the requested clarifications, we have also incorporated additional experimental evidence to strengthen our findings. Notably, we have demonstrated that the proposed architecture not only performs well but also generalizes effectively across diverse dimensions such as flavor profiles and even emotional responses elicited by consuming functional products. This novel aspect underscores the human-centered potential of our approach, bridging technical innovation with meaningful applications in real-world scenarios.
> >
> > We hope these enhancements address any lingering concerns and provide compelling evidence of the robustness and impact of our work. We humbly suggest a reconsideration of the evaluation, as we believe this version significantly advances the original submission in both quality and relevance.
> >
> >
> > In this link, you can view the article where we have highlighted in red not only the observations you provided but also those from others,  It is worth noting that significant portions of the article have been restructured, and new research findings have been discovered:
> >
> > https://openreview.net/pdf?id=B6xUlbgP7j
> >
> > Once again, thank you for your valuable feedback and for considering our appeal. Please let us know if there are additional elements we can further clarify or improve.
> >
> > Warm regards,

---

### Meta-Review · Area_Chair_sfYh · 2024-12-21

**Metareview:**

The paper uses dimensionality reduction methods and DNNs to establish a relationship between EEG signals recorded in the brain and taste preferences, finding that preferences could be predicted by low beta and gamma frequency band activity.  Reviewers felt that although the paper contained some promising ideas, it did not represent a substantial enough methodological or scientific advance to warrant acceptance to ICLR.  I regret that the paper cannot be accepted to this year's meeting, but I wish the authors the best of luck in revising it for publication elsewhere.

**Additional Comments On Reviewer Discussion:**

This paper was a strong reject (1,1,3,3).  The authors posted rebuttals that were somewhat vague and lacking in detail, which did not persuade reviewers to raise their scores.

---

### Decision · Program_Chairs · 2025-01-22

Reject